# MORPHEUS Phase II–III Study: A Pre-Planned Interim Safety Analysis and Preliminary Results

**DOI:** 10.3390/cancers14153665

**Published:** 2022-07-28

**Authors:** Aurelie Garant, Carol-Ann Vasilevsky, Marylise Boutros, Farzin Khosrow-Khavar, Petr Kavan, Hugo Diec, Sylvain Des Groseilliers, Julio Faria, Emery Ferland, Vincent Pelsser, André-Guy Martin, Slobodan Devic, Te Vuong

**Affiliations:** 1Department of Radiation Oncology, UT Southwestern Medical Center, Dallas, TX 75390, USA; aurelie.garant@utsouthwestern.edu; 2Department of Surgery, Jewish General Hospital, McGill University, Montreal, QC H3T1E2, Canada; carol-ann.vasilevsky@mcgill.ca (C.-A.V.); marylise.boutros@mcgill.ca (M.B.); julio.faria@mcgill.ca (J.F.); 3Department of Oncology, McGill University, Montreal, QC H4A3T2, Canada; farzin.khosrow-khavar@mail.mcgill.ca (F.K.-K.); petr.kavan@mcgill.ca (P.K.); 4Department of Surgery, Hospital Pierre Boucher, Montreal, QC J4M2A5, Canada; hugo_diec@hotmail.com (H.D.); sdgmd@videotron.ca (S.D.G.); 5Department of Oncology, Hospital Pierre Boucher, Montreal, QC J4M2A5, Canada; emery.ferland@sympatico.ca; 6Department of Radiology, McGill University, Montreal, QC H3T1E2, Canada; vincent.pelsser.med@ssss.gouv.qc.ca; 7Department of Radiation Oncology, Centre Hospitalier Universitaire de Québec, Université Laval, Quebec City, QC G1R3S3, Canada; andre-guy.martin.med@ssss.gouv.qc.ca; 8Medical Physics Unit, Oncology Department, McGill University, Montreal, QC H3T1E2, Canada; slobodan.devic@mcgill.ca; 9Department of Radiation Oncology, Jewish General Hospital, McGill University, Montreal, QC H3T1E2, Canada

**Keywords:** rectum cancer, watchful waiting, organ preservation, non-operative management, brachytherapy, rectum preservation

## Abstract

**Simple Summary:**

This prospective phase II–III randomized trial explores the value of modern technology for prudent radiation dose escalation in the curative management of patients with rectal cancer. It includes 40 operable patients with locally advanced operable rectal cancer. This manuscript illustrates the planned interim analysis, which was intended to confirm safety, including the absence of unexpected toxicities and the inability to perform salvage surgery. The treatment approach involves pelvic chemoradiotherapy followed by either an additional external beam radiation boost or three weekly boosts of brachytherapy, which are adapted in real-time to tumor response. This proof-of-principle study shows that careful radiotherapy dose escalation leads to high proportions of sustained complete clinical response in patients with rectal cancer with acceptable toxicity. Ongoing accrual to the phase III component of the trial is in progress.

**Abstract:**

Background: We explored image-guided adaptive endorectal brachytherapy patients electing non-operative management for rectal cancer. We present the first pre-planned interim analysis. Methods: In this open-label phase II–III randomized study, patients with operable cT2-3ab N0 M0 rectal cancer received 45 Gy in 25 fractions of pelvic external beam radiotherapy (EBRT) with 5-FU/Capecitabine. They were randomized 1:1 to receive either an EBRT boost of 9 Gy in 5 fractions (Arm A) or three weekly adaptive brachytherapy (IGAEBT) boosts totaling 30 Gy (Arm B). Patient characteristics and toxicity are presented using descriptive analyses; TME-free survival between arms with the intention to treat the population is explored using the Kaplan–Meier method. Results: A total of 40 patients were in this analysis. Baseline characteristics were balanced; acute toxicities were similar. Complete clinical response (cCR) was 50% (*n* = 10/20) in Arm A and 90% in Arm B (*n* = 18/20). Median follow-up was 1.3 years; 2-year TME-free survival was 38.6% (95% CI: 16.5–60.6%) in the EBRT arm and 76.6% (95% CI: 56.1–97.1%) in the IGAEBT arm. Conclusions: Radiation intensification with IGAEBT is feasible. This interim analysis suggests an improvement in TME-free survival when comparing IGAEBT with EBRT, pending confirmation upon completion of this trial.

## 1. Introduction

Rectal cancer is one of the most common malignancies globally, frequently affecting patients above the age of 50 [1]. Colorectal cancer is the third most common cancer and the second leading cause of cancer death globally, with rectal cancer accounting for one-third of these cases [2]. Total Mesorectal Excision (TME) combined with neoadjuvant external beam radiotherapy (EBRT) with/without 5-Fluorouracil (5-FU) chemotherapy (CTRT) remains the standard management of this condition. Following treatment, sexual and urinary dysfunction may occur [3]: in fact, 40% of patients have reduced quality of life (QoL) following colorectal/coloanal anastomosis, largely due to bowel dysfunction commonly referred to as Low Anterior Resection Syndrome (LARS) [4,5]. 

Twenty years ago, Habr-Gama explored the watch-and-wait (W&W) approach [6] for rectal cancer patients who would require a permanent stoma. With an EBRT course of 54 Gy in 30 fractions followed by four cycles of 5-FU, 49% of the patients developed a complete clinical response (cCR). Subsequent reports [7] demonstrated that most local relapses could undergo surgery without a measurable impact on survival [8,9]. Since that time, several groups have conducted research on non-operative management (NOM) [10,11,12,13,14,15]. Emerging data show that dose escalation increases tumor regression and suggests that more patients could benefit from planned NOM [16,17,18,19].

Patients of all ages can undergo radiotherapy with good tolerance. In addition to conventional EBRT, image-guided adaptive endorectal high dose rate brachytherapy (IGAEBT) can complement local therapy by placing an applicator directly at the tumor surface, thus overcoming issues related to natural bowel motion [20,21]. 

We present the interim analysis of a randomized trial exploring IGAEBT as a boosting technique for NOM. In this first interim analysis, we compare toxicity and TME-free survival between IGAEBT and EBRT treatment arms.

## 2. Materials and Methods

### 2.1. Study Design and Participants

This is a randomized phase II–III study of consecutive patients treated between April 2017 and March 2020, approved by our institutional review board and activated in 3 institutions. This trial is registered on clinicaltrials.gov (NCT03051464). Each patient underwent multidisciplinary evaluation by a team of colorectal surgeons, pathologists, radiation oncologists, medical oncologists and radiologists prior to enrollment. MRI was reviewed for each patient prior to recruitment by expert radiologist at Tumour Board. 

Patients were eligible if they had clinical cT2-T3ab N0 M0 by MRI. In order to ensure proper access for endoscopy and IGAEBT, patients were required to have a rectal tumor occupying ≤50% of the luminal circumference and ≤5 cm length, within 10 cm of the anal verge. Patients had to be at least 18 years of age, eligible for chemotherapy, on birth control (if women of childbearing potential), and able to provide written informed consent.

Patients were excluded from the study if they had more locally advanced stages than the above-mentioned, had received previous pelvic radiation, primary tumor extent into the anal canal, or with a prior history of noncompliance to scheduled clinic visits.

Baseline clinical staging included a complete physical examination including digital rectal exam, colonoscopy, biopsy, Carcinogenic Embryonal Antigen, pelvic magnetic resonance imaging (MRI), and computed tomography (CT) scans of the chest, abdomen and pelvis. 

### 2.2. Procedures

Once patients were enrolled, they underwent a 1:1 randomization to receive either EBRT (Arm A) or IGAEBT (Arm B) boost. Randomization was stratified by T stage, size, tumor location and gender. Randomization was conducted centrally at Lady Davis Research Institute by telephone. All patients began therapy with pelvic EBRT with concurrent 5-FU/Capecitabine. Concurrent chemotherapy with capecitabine (825 mg/m^2^ bid on radiation days) or continuous infusion of 5-FU IV at 225 mg/m^2^ was planned in both treatment arms. Patients were simulated for EBRT, with co-registration of diagnostic MRI. A ≥ 6 MV linear accelerator delivered 3D conformal EBRT of 45 Gy in 25 daily fractions based on published data [22]. The clinical target volume (CTV) included the primary tumor with 2 cm expansion as well as the mesorectum, and at-risk nodal basins seen on MRI/CT, with the perirectal, presacral and internal iliac nodes up to the L5-S1 junction. The posterior obturator nodes were included for tumors <10 cm from the anal verge. 

Subsequently, patients received a boost of radiotherapy for the primary tumor. In Arm A, patients received 9 Gy in 5 fractions to boost PTV (primary gross tumor volume + 2 cm expansion + 1 cm planning margin) with concomitant oral capecitabine (Xeloda) or intravenous 5-fluorouracil (5-FU). In Arm B, patients had 3-week break prior to the administration of 30 Gy in 3 weekly fractions, to allow for optimal patient tolerance to the endorectal applicator. A post-EBRT pelvic MRI was acquired, and a gastroenterologist placed radio-opaque clips at the edge of the residual tumor. We delivered treatment with an eight-channel intracavitary applicator and a microelectronic HDR brachytherapy Ir-192 afterloader (Nucletron/Elekta; Veenendaal, The Netherlands). The weekly clinical target volume (HDR-CTV) delineation accounted for tumor regression during the course of radiotherapy, and we proceeded with adaptive planning for each IGAEBT session using a new CT simulation for each fraction [21]. 

Post-treatment monitoring included clinical evaluation with endoscopic evaluation 13 weeks after completion of radiotherapy. Subsequently, for patients achieving complete clinical response (cCR), clinical and endoscopic evaluations were obtained every 3 months for the first three years then every 6 months until the end of the fifth year; in non-complete clinical responders (ncCR), TME was indicated, and post-TME endoscopic scheduling was adjusted to every 6 months for the first two years then annually until the end of the fifth year. For cCR patients, an MRI was obtained every 6 months for the first 2 years and then yearly until year 5; MRIs were discontinued in ncCR patients. CT scans of the chest, abdomen, and pelvis were carried out annually for five years for all patients.

Quality of life (QoL) data (LARS questionnaire and EORTC QLQ-CR30) were acquired every 6 months for the first two years then annually until the end of the fifth year. Their analysis was not mandated in this early analysis by the IDMC. 

### 2.3. Outcomes

The primary outcome of this proposal was TME-free survival defined as time from date of randomization to either TME or death in the intention to treat population. Rectum preservation was assessed by the investigations performed during the clinical evaluation (at 13 weeks from end of treatment) and monitored during follow-up. Secondary outcomes including local recurrence, disease-free survival, overall survival, QoL, and a composite endpoint consisting of organ preservation without locoregional (rectal–pelvic) recurrence are planned to be reported at the time of completion of the phase III study.

The following 3 criteria were required to achieve cCR: no mass remaining upon rectoscopic examination, soft palpation on the digital exam, and absence of residual tumor on MRI [23]. If there was interval regression on two consecutive endoscopic evaluations, small soft ulcers were considered acceptable “near complete response” [16,24,25] and eligible for continued surveillance. However, if the 3 criteria specified above were not fulfilled, patients were characterized as non-complete clinical responders. Patient requiring TME is considered as local failure (Non-TME-free) unlike patients salved by TEM or local excision.

Local control was defined as a patient achieving cCR with no subsequent relapse. Persistent disease after treatment was documented as a failure. Local relapses included rectal regrowth after cCR, in-field recurrence after surgery and/or pelvic nodal relapses. Toxicities were recorded based on the CTCAE v4.0 nomenclature.

This planned interim analysis was scheduled upon accrual of 40 patients having completed the trial treatment schedule. Although the initial stopping rules pertained to intervention efficacy, an additional purpose of this data review was to uncover potential excess toxicities and to identify non-salvageable local recurrence above 15% in any arm after cCR, which would preclude consideration of trial expansion in the future. 

### 2.4. Statistical Analysis

This randomized controlled trial is designed as a phase II–III trial with two pre-planned interim analyses. A total of 146 patients are planned for enrollment and assigned 1:1 to either EBRT or IGAEBT. Overall, sample size calculation was conducted using *a priori* hypothesis about the effect size. With a pre-specified conservative absolute difference of 20% in TME-Free Survival between IGAEBT and EBRT arms at two years (50% vs. 30%) corresponding to a hazard ratio of 0.576, 146 patients provide 80% power for a two-sided log-rank test using 5% false positive rate and accounting for 10% loss to follow-up in each treatment arm. This calculation was based on an O’Brien Fleming α-spending function for efficacy boundaries in a group sequential design using two interim analyses which maintain the overall type I error rate for the trial [26,27,28]. The study continued as planned after this first interim analysis as the *p*-value for a two-sided log-rank test at an information fraction of 18% (16 events) did not cross the efficacy boundary (*p*-value: <0.0001). The Independent Data Monitoring Committee (IDMC) reviewed both efficacy and safety results and reported that the efficacy boundary for the primary analysis had not been met and due to no safety concerns, the trial is continuing as planned.

We used descriptive statistics to summarize the baseline characteristics of the study population by trial arm including mean, standard deviation, median, and interquartile range for continuous variables and counts and frequencies for categorical variables. The Kaplan–Meier method and log-rank test were used to compare TME-free survival between treatment arms. The cumulative hazard was estimated using the Nelson–Aalen estimator. We used a negative binomial regression model to calculate incidence rates (per 100 person-years) and corresponding 95% confidence intervals for the primary outcome. Cox proportional hazards models were used to estimate hazard ratios and 95% confidence intervals comparing the rate of composite outcome (TME or death) for patients who were randomized to treatment with IGAEBT in comparison with patients randomized to treatment with EBRT using intention to treat analysis. We conducted sensitivity analyses adjusting for *a priori*-defined prognostic factors assessed at baseline [29,30]. All reported *p*-values are two-sided. Analyses were conducted using the SAS software version 9.4 (SAS Institute Inc., Cary, NC, USA) and R statistical software (Vienna, Austria). 

## 3. Results

In this trial, 40 patients were randomized and are presented in a CONSORT diagram (Figure 1). Baseline characteristics were well balanced in terms of age, tumor location, T stage and tumor size (Table 1). 

All patients proceeded with the treatment assigned at randomization. The acute treatment-related toxicities were similar (Table 2). Two patients in Arm B obtained a grade 3 proctitis (10%) which is lower than the Hebert trial [31]. At the time of this analysis, two patients had died in the Arm B group. One patient in the Arm B group was admitted during week 2 for acute congestive heart failure and despite intensive care, his condition was uncontrolled and he died in week 8. A second patient in the Arm B group was declared ncCR and died from an acute myocardial infarction (MI) while awaiting his salvage surgery. He was known for a previous MI and had diabetes. Both were qualified as cancer deaths. In the Arm A group, a patient had a persistent pelvis abscess and required numerous surgical interventions. In the Arm B group, two patients had grade 3 proctitis with symptomatic rectal ulcers; one patient required transfusion, and both received plasma argon therapy that healed after 5 and 6 months from treatment completion, with one of the patients requiring hyperbaric oxygen therapy.

The proportion of patients ever achieving complete clinical response over the course of this study was 50% (*n* = 10/20) in Arm A and 90% in Arm B (*n* = 18/20). The median follow-up (interquartile range) for the primary outcome was 1.3 years (IQR: 0.8–2.6) and 2.4 years (IQR: 1.4–3.4) for toxicity outcomes. Two patients, one from each arm with tumor regrowth following cCR, refused TME surgery and underwent an uncomplicated transanal endoscopic microsurgery salvage resection: both currently have no evidence of disease. The post-operative complication distribution for patients undergoing TME is also shown in Table 3. The mean hospital length of stay for TME patients was 9 and 8 days, respectively, for Arm A and B with no post-operative deaths recorded within the first 30 days. The distribution of complications between the two arms is similar. One patient in Arm A was deemed unresectable at the time of surgery on the basis of more extensive disease than predicted by imaging and dense fibrosis, while none were found to be unresectable in Arm B. Although numbers are limited at this time, surgeons reported more fibrosis in the surrounding tissues in arm A (*n* = 2) that required longer dissection than in Arm B (*n* = 0). 

Overall, patients in the IGAEBT arm had higher TME-free survival in comparison with the EBRT arm (log-rank *p*-value = 0.006, Figure 2). However, interim efficacy boundaries were not crossed using O’Brien Fleming alpha-spending function. TME-free survival at two years was 38.6% (95% CI: 16.5–60.6%) in Arm A and 76.6% (95% CI: 56.1–97.1%) in Arm B (Figure 2). The rate of the primary endpoint (95% CI) comparing IGAEBT with EBRT per 100 person-years was 4.6 (1.1–20.3) vs. 55.4 (22.6–135.8) within the first year of follow-up and 4.5 (1.1–17.8) vs. 11.3 (2.8–45.3) after the first year of follow-up. The corresponding plot of cumulative hazard for TME or death is shown in Appendix A. Overall, the HR for the composite endpoint of TME or surgery comparing IGAEBT with EBRT was 0.23 (95% CI: 0.07–0.71, *p*-value: 0.01) (Table 4). In sensitivity analysis adjusting for *a priori*-defined prognostic factors, the HR was 0.13 (95% CI: 0.03–0.50, *p*-value: <0.01) (Table 4).

## 4. Discussion

In this interim analysis of a randomized phase II/III trial, designed initially to be the third arm of the OPERA trial, we observed that endoluminal radiotherapy dose escalation using IGAEBT offers a favorable safety profile in patients presenting with curable, stage I–IIA rectal cancer (AJCC 8th edition staging). Patient and tumor characteristics are shown in Table 1 with no significant difference between the two arms. Most of our tumor population were T3 and 70 % were sized from 3 to 5 cm. There was no major difference in grade 3 acute toxicity between the two arms, as shown in Table 2. 

The proportion of complete clinical response was 50% (*n* = 10/20) in Arm A and 90% in Arm B (*n* = 18/20). A total of 12 and 4 patients, respectively, in Arms A and B underwent TME surgery or died (Table 4); length of stay was 9 and 8 days, respectively, for Arms A and B with no significant difference in post-operative surgical complications (presacral bleeding, pelvic abscess, anastomotic leak, surgical re-intervention) between the two arms as well as in the medical complications within the first month after surgery. One patient in the Arm A group was deemed unresectable because of pelvic fibrosis. No death was observed within the first 30 days after surgery. Two patients, one from each arm group with tumor regrowth, refused TME surgery and underwent an uncomplicated salvage transanal endoscopic microsurgery (TEM) and both (2 (11%) out of 18 patients requiring salvage surgery) are currently with no evidence of disease at 9 and 36 months follow-up post-intervention. Although full trial details are pending publication, our surgical salvage and post-operative complication data are in keeping with those reported in the OPERA multicenter European phase III organ preservation trial [32]. TEM/local excision may be an interesting option to consider and was also reported in 21 out of 49 patients (42.8%) requiring salvage surgery in the OPERA trial as the International W&W database with 1009 patients showed that 97% of tumor regrowth is within the rectal wall [15]. At this early evaluation step, with 76.6% and 38.6% TME-free survival rates observed in Arm B and A, respectively, (log-rank *p* = 0.001), our data are in favor of dose escalation with an IGAEBT boost (Figure 2). We acknowledge that early outcomes must be considered with caution, especially given that post-treatment endoscopic evaluation occurred at 19 weeks post-initiation of radiotherapy in Arm A, versus 24 weeks in Arm B. This requires confirmation in the completed trial as per our (IDMC). 

The results of our randomized phase II–III study are further supported by recent literature reviews [31,33], which suggest that a higher radiotherapy dose may be required to eradicate cT3-4 disease than cT1-2 tumors. An updated analysis of the Danish brachytherapy boost trial also confirms that endoluminal therapy provides a high proportion of organ preservation, noting 69% without local regrowth among cCR patients at 5 years [34]. Our analysis suggests the benefits of dose escalation and supports ongoing efforts to counsel patients on planned NOM with adaptive radiation planning [35], rather than an opportunistic approach [36] with traditional EBRT dosage. 

Of note, two previous randomized trials had failed to demonstrate a benefit to dose escalation in patients with locally advanced rectal cancer [37,38], but the Lyon R96-02 (ref. [28]) demonstrated that an endocavitary boost significantly increase cCR and organ preservation. These two phase II trials opted for a primary endpoint of pCR, with an interval from the end of treatment to surgery ranging from 8–12 weeks. In our opinion, our trial differs from these previous works for multiple reasons. One main factor is our extended time between treatment and response evaluation: indeed, our first post-treatment evaluation occurred at 13 weeks, and near cCR patients were allowed to continue with additional surveillance, in light of the possibility of achieving cCR in subsequent months. Another factor is the adaptive component of our boost technique, which occurs at the end of the radiotherapy course and takes into account any interval downsizing with direct placement of the applicator onto the tumor. The technical challenge of this technique remains in tumor delineation after EBRT. The residual tumor is not visible on CT simulation and does require the aid of direct visualization by endoscopy and tumor markers during the weekly session. This differs from the RECTAL-BOOST, where the boost was delivered at the start of the course as an EBRT technique with moderately sized margins; likewise, in the Danish–Canadian trial, two focal HDR brachytherapy boosts were scheduled within a planned EBRT course, where EBRT would not be given on HDR days.

In addition to the above, the HERBERT feasibility trial had previously explored HDR brachytherapy boost in elderly patients with rectal cancer [39,40]. This trial differs greatly from our current research, given a different patient population with higher median age and comorbidity profile precluding surgery; moreover, several technical factors surrounding the delivery of HDR brachytherapy were different from our current study. The technique used in the HERBERT study was derived from previous works on brachytherapy in the preoperative setting [41], where adaptation was not considered, given plans to move forward with TME. In our opinion, the adaptive real-time boost planning of each brachytherapy fraction, as well as the supplemental shielding techniques employed in the current study provided a more optimal therapeutic index, to allow for maximal response and diminished risk of normal mucosal toxicity in the non-operative setting.

Our study provides important information regarding the safety of salvage surgery in patients not in cCR or with tumor regrowth. We previously reported our initial favorable NOM clinical experience using endorectal dose escalation brachytherapy in the elderly, unfit population in whom surgery was not an option [20]. IGAEBT allows for safe salvage surgery in the NOM management of patients with rectal cancer [34]. Interestingly, surgeons reported more fibrosis in the surrounding tissues in arm A that required longer dissection than in arm B. Furthermore, our interim results confirm the hypotheses suggested by Hall et al. [18] and Appelt et al. [33] on the benefits of dose escalation using IGAEBT. 

Some of the limitations of our interim analysis include small sample size, limited follow-up, and limited ability to make definitive conclusions until the final analysis is complete. We acknowledge that there may be ethical considerations in enrolling further patients in this trial, given a strong signal favoring the experimental Arm B for planned NOM.

## 5. Conclusions

Our study shows that targeted dose escalation with the endoluminal brachytherapy technique (IGAEBT) allows safe salvage surgery to occur in the NOM management of patients with rectal cancer. Organ function outcomes results from the near future analysis are needed. Should it prove to be favorable along with the present toxicity profile and the achievement of a high cCR, the role of IGAEBT in the NOM management of this patient population will be established. Accrual of patients in this trial and continued follow-up are ongoing as planned previously. 

## Figures and Tables

**Figure 1 cancers-14-03665-f001:**
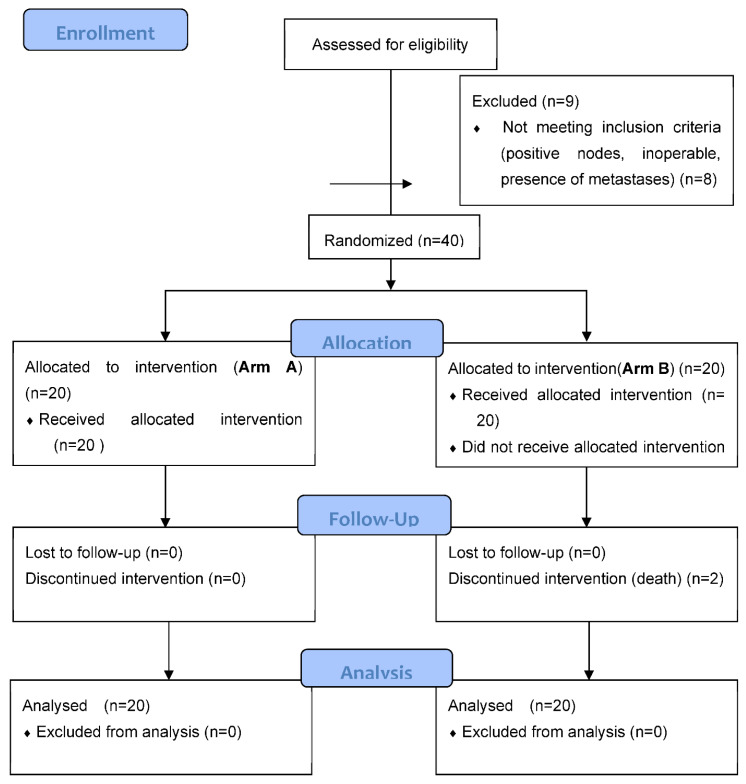
CONSORT Diagram.

**Figure 2 cancers-14-03665-f002:**
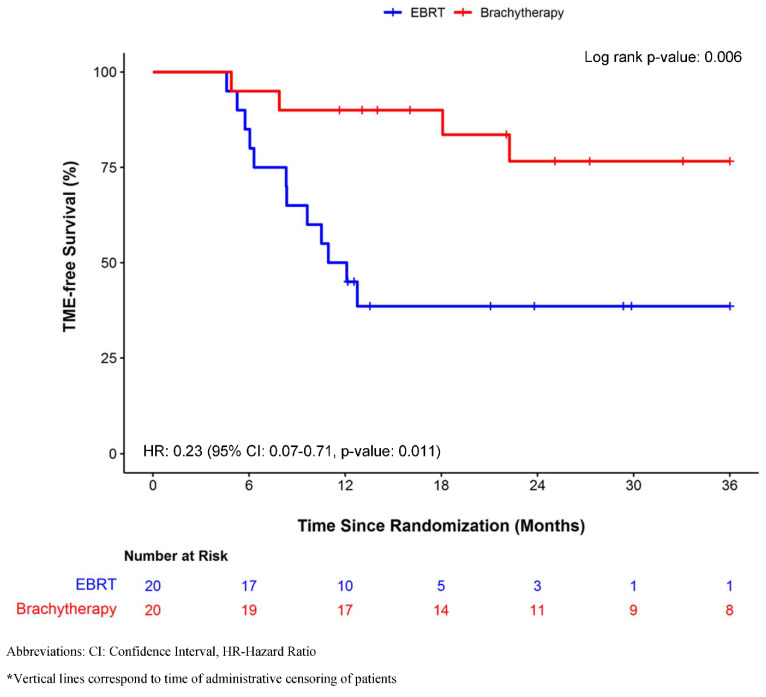
TME-free Survival probability when comparing IGAEBT versus EBRT in the intention to treat population.

**Table 1 cancers-14-03665-t001:** Patient and tumor characteristics by study arm.

	EBRT (*n* = 20)	IGAEBT (*n* = 20)
**Age (median, range)**	67.5 (42–89)	64.5 (46–87)
**Age (mean, std)**	69.7 (12.2)	68.0 (12.0)
**Male Sex, *n* (%)**	15 (75)	12 (60)
**Tumor stage, *n* (%)**		
cT2	8 (40)	6 (30)
cT3	12 (60)	14 (70)
cN0	20 (100)	20 (100)
**Tumor location, *n* (%)**		
Mid-third (5 to 10 cm from anal verge)	9 (45)	7 (35)
Lower-third (5 cm from anal verge)	11 (55)	13 (65)
**Tumor length on sagittal MRI view, *n* (%)**		
<3 cm	6 (30)	6 (30)
≥3 cm	14 (70)	14 (70)
**Pathology, *n* (%)**		
Well differentiated	8 (40)	8 (40)
Moderately differentiated	9 (45)	8 (40)
Not specified	3 (15)	4 (20)

**Table 2 cancers-14-03665-t002:** Acute radiation-related and chemotherapy-related toxicity outcomes by trial arm.

Toxicities	EBRT	IGAEBT
	(*n* = 20)	(*n* = 20)
**Chemoradiotherapy Toxicities**	18 (90)	15 (75)
**Anemia, *n* (%)**	8 (40)	7 (35)
G1	5 (25)	5 (25)
G2	3 (15)	2 (10)
≥G3	0 (0)	0 (0)
**Leukopenia, *n* (%)**	2 (10)	4 (20)
G1	0 (0)	2 (10)
G2	2 (10)	1 (5)
≥G3	0 (0)	1 (5)
**Diarrhea, *n* (%)**	17 (85)	10 (50)
G1	9 (45)	7 (35)
G2	8 (40)	3 (15)
≥G3	0 (0)	0 (0)
**Radio-dermatitis, *n* (%)**	16 (80)	12 (60)
G1	2 (10)	3 (15)
G2	13 (65)	8 (40)
≥G3	1 (5)	1 (5)
**IGAEBT Toxicities**		
**Proctitis, *n* (%)**	NA	18 (90)
G1	NA	13 (65)
G2	NA	3 (15)
≥G3	NA	2 (10)
**Death, *n* (%)**	0 (0)	2 (10)

**Table 3 cancers-14-03665-t003:** Acute surgery-related toxicity outcomes by trial arm among patients who received surgery (n corresponds to number of events).

Outcome	EBRT (*n* = 12 pts)	IGAEBT (*n* = 2 pts)
Urinary tract trauma infection	4	0
Abdominal wall infection	1	1
Abscess	3	1
Delirium	1	0
Presacral bleeding	1	1
Rectal prolapse	1	0
Anastomotic leak	1	0
Cardiopulmonary events	1	0
Small bowel obstruction	1	0
Post-surgery arm neuropathy	1	0
COPD exacerbation	1	0
Urinary infection trauma	2	0
Ongoing intra-abdominal abscess	1	0
Right arm neuropathy	1	0
Re-operation	2	0

Abbreviations: COPD–Chronic obstructive pulmonary disease. This table only applies to TME surgery.

**Table 4 cancers-14-03665-t004:** Hazard ratio and 95% confidence interval of composite outcome (TME or death) when comparing IGAEBT versus EBRT arms in the intention to treat population.

Treatment	Patients	Events	Person-Years	Incidence Rate (95% CI) ^a^	HR (95% CI)	*p*-Value	Adjusted HR (95% CI) ^b^	*p*-Value
EBRT	20	12	24.0	61.1 (27.7–134.8)	Ref		Ref	
IGAEBT	20	4	46.9	9.2 (3.2–26.4)	0.23 (0.07–0.71)	0.011	0.13 (0.03–0.50)	0.003

Abbreviations: HR-hazard ratio. ^a^. Per 100 person-years, estimated using negative binomial regression. ^b^. Adjusted for age, gender, tumor stage, tumor location, tumor size, differentiation.

## Data Availability

Data available on request due to ethical restrictions.

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
