# Peer review of "MORPHEUS Phase II–III Study: A Pre-Planned Interim Safety Analysis and Preliminary Results"

_cancers, 2022, doi:10.3390/cancers14153665_

Round 1

Reviewer 1 Report

This is an excellent paper reporting the palnned interim analysis of a phase II-III trial

The results are excellent for safety and  very encouraging for the phase III which should be positive in favor of the endoluminal boost.

Some minor comments :

May be mention that this trial was designed to be matching the OPERA trial

Abstract :  Complete clinical response : May be it is cCR whic is Clinical complete resonse

MRI : is there any expert review of the MRI diagnosis ?  Is there a distinction between T3 a, b  and c , d ?

Local control : Is a patient with no pelvic tumor after TME considered as local control : yes or No ?

Results : NCCR  should probably be ncCR

Table 3    IGAEBT : 2 and 0 in the column ? 

NOM : may be explain that it means no TME but it is not including TEM/ local excision

Discussion : two randomized trias failed to demonstrate benefit of a boost…. May be mention that the Lyon R96-02 trial (ref 28) demonstarted that an endocaitary boost significantly increase safely cCR and organ preservation.

Deliberate NOM or Planned NOM (ref 36)

Author Response

Comment 1: May be mention that this trial was designed to be matching the OPERA trial

Reply: In the sentence starting with: “In this interim analysis of a randomized phase II/III trial, …” we added: “ … , designed initially to be the third arm of the OPERA trial, …”

Comment 2: Abstract : Complete clinical response : May be it is cCR which is Clinical complete response

Reply: c CR is Clinical complete response

Comment 3: MRI : is there any expert review of the MRI diagnosis ? Is there a distinction between T3 a, b and c , d ?

Reply: At the end of paragraph starting with: “This is a randomized phase II-III study of consecutive patients …” we added a sentence: “MRI was reviewed for each patient prior to recruitment by expert radiologist at Tumour Board.”

Comment 4: Local control : Is a patient with no pelvic tumor after TME considered as local control : yes or No ?

Reply: No

Comment 5: Results : NCCR should probably be ncCR

Reply: We changed it throughout the text

Comment 6: Table 3 IGAEBT : 2 and 0 in the column ?

Reply: We modified Table 3

Comment 7: NOM : may be explain that it means no TME but it is not including TEM/ local excision

Reply: At the end of paragraph starting with: “The following 3 criteria were required to achieve cCR …” we added a sentence: “Patient requiring TME is considered as local failure ( Non TME free) unlike patients salvaged by TEM or local excision.”

Comment 8: Discussion : two randomized trials failed to demonstrate benefit of a boost…. May be mention that the Lyon R96-02 trial (ref 28) demonstrated that an endocavitary boost significantly increase safely cCR and organ preservation.

Reply: At the end of the sentence starting with: “Of note, two previous randomized trials had failed to demonstrate a benefit to dose escalation …” we added a text: “…but the Lyon R96-02 (ref 28) demonstrated that an endocavitary boost significantly in-crease cCR and organ preservation.”

Comment 9: Deliberate NOM or Planned NOM (ref 36)?

Reply: It is planned NOM – we changed it throughout the text.

Reviewer 2 Report

Many thanks for giving me the opportunity to review this manuscript. In the present randomized phase II/III trial the authors assessed the role of endoluminal brachytherapy boost as compared to EBRT boost after standard CRT for patients with cT2-3a/bN0 rectal cancer. The authors report a superior TME-free survival and organ preservation rates in favor of brachytherapy. I congratulate the investigators on completing this important trial.Altogether, this is a well-written report of the preliminary data of the MORPHEUS trial that merits publication. The methods and results are sound, the discussion is comprehensive and up-to-date, and the take home message is clear.  Some minor points could be addressed as follows: 

-Can you please comment a bit more on the toxicity after brachy, and how does it compare to the literature data? 

-Are there any data on QoL/PROMs already available in this preliminary analysis? 

-Can you briefly touch upon the technical challenges of brachytherapy, especially for readers unfamiliar with this topic? 

Author Response

Comment 1: Can you please comment a bit more on the toxicity after brachy, and how does it compare to the literature data?

Reply: After the sentence: “The acute treatment related toxicities were similar (Table 2).” We added a sentence: “Two patients in arm B got a grade 3 proctitis (10%) which is lower than Hebert trial [34].”

Comment 2: Are there any data on QoL/PROMs already available in this preliminary analysis?

Reply: Yes, they were collected but not analyzed for this early interim review as per IDMC. After the sentence starting with: “Quality of life (QoL) data (LARS questionnaire and EORTC QLQ-CR30) was acquired every 6 months …” we added a sentence: “Their analysis was not mandated in this early analysis by the IDMC.”

Comment 3: Can you briefly touch upon the technical challenges of brachytherapy, especially for readers unfamiliar with this topic?

Reply: After the sentence: “The technical challenge of this technique remains in tumor delineation after EBRT.” We added a sentence: “ The residual tumor is not visible on CT simulation and does require the aids of direct visualization by endoscopy and tumor markers during the weekly session.”